# Bacteriophages for Controlling *Staphylococcus* spp. Pathogens on Dairy Cattle Farms: In Vitro Assessment

**DOI:** 10.3390/ani14050683

**Published:** 2024-02-22

**Authors:** Ewelina Pyzik, Renata Urban-Chmiel, Łukasz Kurek, Klaudia Herman, Rafał Stachura, Agnieszka Marek

**Affiliations:** 1Department of Veterinary Prevention and Avian Diseases, Faculty of Veterinary Medicine, University of Life Sciences in Lublin, 20-033 Lublin, Poland; ewelina.pyzik@up.lublin.pl (E.P.); klaudia.herman@up.lublin.pl (K.H.); agnieszka.marek@up.lublin.pl (A.M.); 2Department and Clinic of Animal Internal Diseases, Sub-Department of Internal Diseases of Farm Animals and Horses, Faculty of Veterinary Medicine, University of Life Sciences in Lublin, 20-033 Lublin, Poland; lukasz.kurek@up.lublin.pl; 3Agromarina Company, Kulczyn-Kolonia 48, 22-235 Hańsk Pierwszy, Poland; agromarina@wp.pl

**Keywords:** bacteriophages, cows, lameness, *Staphylococcus* spp.

## Abstract

**Simple Summary:**

The aim of this in vitro study was to evaluate the antibacterial effectiveness of bacteriophages specific for *Staphylococcus* spp. isolated from dairy cattle as potential tools for maintaining environmental homeostasis. The material for this study was 56 samples obtained from 49 dairy cows with limb and hoof infections. Phages were isolated from cattle housing systems—litter and floor samples. Phage activity (plaque forming units, PFU/mL) was determined on double-layer agar plates. Phage morphology was examined using TEM microscopy. Molecular characteristics were determined using PCR. In total, 52 strains classified as *Staphylococcus* spp. were isolated, of which 16 were used as hosts for specific bacteriophages. Nearly all isolates (94%) showed resistance to neomycin, and 87% of strains were resistant to spectinomycin. We obtained 14 bacteriophages, but only three, with the broadest spectrum of antibacterial activity and stable lytic titres, were used for further analysis. This study showed a wide spectrum of activity of bacteriophages against *Staphylococcus* spp. strains isolated from cattle with lesions in their limbs and hooves.

**Abstract:**

Pathogenic *Staphylococcus* spp. strains are significant agents involved in mastitis and in skin and limb infections in dairy cattle. The aim of this study was to assess the antibacterial effectiveness of bacteriophages isolated from dairy cattle housing as potential tools for maintaining environmental homeostasis. The research will contribute to the use of phages as alternatives to antibiotics. The material was 56 samples obtained from dairy cows with signs of limb and hoof injuries. *Staphylococcus* species were identified by phenotypic, MALDI-TOF MS and PCR methods. Antibiotic resistance was determined by the disc diffusion method. Phages were isolated from cattle housing systems. Phage activity (plaque forming units, PFU/mL) was determined on double-layer agar plates. Morphology was examined using TEM microscopy, and molecular characteristics were determined with PCR. Among 52 strains of *Staphylococcus* spp., 16 were used as hosts for bacteriophages. Nearly all isolates (94%, 15/16) showed resistance to neomycin, and 87% were resistant to spectinomycin. Cefuroxime and vancomycin were the most effective antibiotics. On the basis of their morphology, bacteriophages were identified as class *Caudoviricetes*, formerly *Caudovirales*, families *Myoviridae*-like (6), and *Siphoviridae*-like (9). Three bacteriophages of the family *Myoviridae*-like, with the broadest spectrum of activity, were used for further analysis. This study showed a wide spectrum of activity against the *Staphylococcus* spp. strains tested. The positive results indicate that bacteriophages can be used to improve the welfare of cattle.

## 1. Introduction

Infections induced by pathogenic strains of *Staphylococcus* spp. in cattle are a significant health problem in animal herds and pose an unquestionable risk to humans [1]. Methicillin-resistant *Staphylococcus aureus* (MRSA) bacteria, which are responsible for mastitis infections, are particularly dangerous. MRSA strains and other *Staphylococcus* species are also isolated from skin and limb infections and even directly from animal housing environments [1,2,3].

*Staphylococcus*, as one of the most common saprophytic bacteria present on limbs, can enhance hoof inflammation and is most often isolated in these lesions on cows [4,5]. According to the authors cited, the presence of these bacteria in limb and hoof lesions creates an ideal environment for the colonization and multiplication of bacteria, including *Fusobacterium necrophorum*, *Treponema* spp., and others. Lameness in dairy cows has an economic impact on production, in part by reducing milk production by about 10%, with a 360 kg reduction in milk yield over 305 days of lactation, and by affecting fertility in cows, reducing pregnancy rates by 25% compared to non-lame cows [6]. The average total costs of lameness related to reduced milk production, reduced reproductive performance, culling costs, and recurrence of lameness are 20.9%, 23.6%, 27.6%, and 7.6% of the total costs, respectively [7].

A major threat to the effective control of infections induced by pathogenic *Staphylococcus* strains is the widespread phenomenon of multidrug resistance. Many studies have shown that the main reservoir of drug-resistant *Staphylococcus* spp. strains is livestock, from which even commensal strains showing multidrug resistance (MDR) to at least one antibiotic from at least three groups of relevant antibiotics have been isolated [8,9,10,11]. A significant threat is the wide spread of *Staphylococcus* spp. strains, which are often sensitive to only one group of antibiotics [11,12]. Many studies on the drug resistance of *Staphylococcus* spp. strains have found that most of them (over 70%) had the ability to produce a biofilm, which significantly increased drug resistance to various groups of antibiotics [13,14].

Another problem with the use of antibiotics to fight infections is the presence of their residues in various products of animal origin, which not only requires a withdrawal period, but above all, raises serious health concerns among consumers regarding products of animal origin [15]. Major challenges also arise from legislative changes in the EU imposing strict control of the use of antibiotics, as well as a total ban on the use of certain groups of antibiotics in veterinary medicine [16].

This situation necessitates a search for alternatives to antibiotics for fighting infections in cattle caused by *Staphylococcus* spp. One proposed solution is the use of bacteriophages as an alternative to antibiotics in the control of infections.

Bacteriophages (bacterial viruses) show a specific affinity for individual types of bacteria. They are the most abundant biological form on earth (10^32^ virions) and are present in diverse environments (wastewater, soil, food products, animals, and humans). The presence of bacteriophages is a natural mechanism, ensuring the proper balance of different bacteria in the natural environment [17]. Bacteriophages can be lytic or lysogenic (the integration of phage DNA with bacterial nucleic acid). For phage therapy, however, only strongly lytic phages should be used to avoid the transfer of resistance genes between bacteria [18]. Bacteriophages are used in the experimental treatment of various bacterial infections in humans and animals, with high confirmed rates of elimination of up to 99–100% of pathogens [19].

Research [20,21] indicates that experimental bacteriophage therapies have shown promising results not only in controlling bacterial pathogens but also in preventing infections. The effectiveness of phage therapies in combating *Staphylococcus* spp. infections in humans and animals has been demonstrated in numerous experiments, including our own in vitro study [22].

When considering the potential use of bacteriophages as alternative antimicrobial agents to antibiotics, it seems very much justified to undertake research to assess the effectiveness of bacteriophages isolated from animal housing environments in controlling bacteria causing infections in cattle. Therefore, the aim of the present study was to assess the effectiveness of bacteriophages specific for the pathogenic strains of *Staphylococcus* spp. isolated from limb and hoof injuries as predisposing factors to lameness in dairy cattle.

## 2. Materials and Methods

The material for this study—both environmental samples and samples obtained from cows—was collected from five dairy farms located in the Lublin region (Poland). Five samples for bacteriophage isolation were collected from the environment (litter, floor) of each farm. One floor sample was collected from each farm.

The material for this study consisted of 56 swabs collected from 49 Holstein–Friesian (HF) dairy cows with varying degrees of advanced limb infections (seven cows had two legs with infections) in their third and fourth lactation cycles, with clinical signs of inflammation of the hoof horn or hoof area. The material from cows with lameness was obtained only once during their first veterinary examination by swabbing the skin in the interdigital space of the hind limb or another part of the injured limb. In seven cases, the hooves of both hind limbs showed signs of infection, so samples were taken from both the left and right hind limb. Detailed information is presented in Table 1.

Directly after the material was collected onto transport substrates, it was transported to the laboratory, where in sterile conditions 5 mL of TSB was added to each sample, followed by incubation in a water bath at 37 °C for 18 h. The resulting broth cultures were then purified in Petri dishes using 5% agar with sheep blood and Chapman selective and differential media (Figure 1).

### 2.1. Isolation of Staphylococcus *spp.* Strains

Bacteria were isolated on two types of media, i.e., mannitol agar (Chapman medium, BTL, PL) and Columbia agar with 5% sheep blood (BTL, PL), in aerobic conditions at 37 °C for 24 h. The cultures were incubated in TSB (BTL, PL) at 37 °C for 24 h to obtain the optimal growth of pure strains. Phenotypic identification of *Staphylococcus* spp. was carried out by Gram staining and with API STAPH commercial biochemical tests [23].

Species identification was carried out using MALDI-TOF mass spectrometry [23,24], using the UltrafleXtreme MALDI-TOF mass spectrometer (Bruker Daltonics, Bremen, Germany) with a 1000 Hz neodymium-doped yttrium aluminium garnet (Nd:YAG) laser. To this end, individual bacterial colonies grown on Chapman agar were collected from the plates and suspended in 1.2 mL of 75% ethanol solution and then centrifuged at 13,000× *g*/2 min at 20 °C. The supernatant was removed, and the remaining pellet of bacterial cells was extracted by adding 50 μL of formic acid (Sigma-Aldrich, Poznań, Poland) and 50 μL of acetonitrile (Sigma-Aldrich, Poland). After centrifuging, each sample was applied to a marked spot on the stainless steel MTP 384 MALDI AnchorChip TF target plate (Bruker, Germany). Then, 1 μL of matrix solution containing 10 mg/mL HCCA (alpha-cyano-4-hydroxycinnamic acid, Sigma-Aldrich, Poland), separated in 50% acetonitrile and 2.5% TFA (trifluoroacetic acid, Sigma-Aldrich, Poland) and air-dried, was applied to the samples. Then, the MALDI plate was placed in the spectrometer for the automated measurement and interpretation of the data. Prior to the analyses, calibration was performed with a bacterial test standard (Bruker, Germany) containing an extract of *Escherichia coli* DH5 alpha. Mass spectra were processed using MALDI Biotyper 3.0 software (Bruker, Germany), containing 3995 reference spectra corresponding to various types of bacteria.

The results were presented as the 10 best identification matches in a confidence interval from 0.00 to 3.00. The results were interpreted according to the manufacturer’s criteria: a log(score) below 1.700 does not allow for reliable identification; a log(score) between 1.700 and 1.999 enables probable identification at the genus level; a log(score) between 2.000 and 2.299 means secure identification at the genus level and probable identification at the species level; and a log(score) higher than 2.300 (2.300–3.000) indicates highly probable identification at the species level.

### 2.2. Identification of Staphylococcus *spp.* Strains by Multiplex PCR and Antibiotic Resistance Profiles

Genetic material was isolated from the strains using the Novabeads Bacterial DNA kit (Novazym Polska, Poznań, Poland) according to the manufacturer’s protocol. The purity and total content of DNA were determined in ng/mL using the NanoDrop^TM^ Lite Spectrophotometer (Thermo Fisher Scientific, Warszawa, Poland) and separation in a 2% agarose gel at 100 V Bio-Rad Mini-protean electrophoresis system (Bio-Rad Company-Poland, Warszawa, Poland).

The genetic material was identified using multiplex PCR using primer pairs specific for the *nuc* sequence characteristic of *Staphylococcus aureus* and the *mec*A gene determining resistance to methicillin.

The reaction was performed using primers for the genes *nuc*1 with the sequence 5′-GCGATTGATGGTGATACGGTT-3′ (melting temp. 52.4 °C), *nuc*2 with the sequence 5′-AGCCAAGCCTTGACGAACTAAAGC-3′ (melting temp. 57.4 °C), *mec*A1 with the sequence 5′-AAAATCGATGGTAAAGGTTGGC-3′ (melting temp. 51.1 °C), and *mec*A2 with the sequence 5′-AGTTCTGGCTACCGGATTTGC-3′ (melting temp. 57.1 °C). The PCR reaction mixture was prepared in Eppendorf tubes in a total amount of 25 μL (RNase-free distilled water 12.5 μL, buffer 2.5 μL, MgCl_2_ 2 μL, dNTP 0.8 μL, Taq polymerase (5 U/µL) 0.2 μL, Primer nuc1 1 μL, Primer nuc2 1 μL, Primer mecA1 1 μL, Starter mecA2 1 μL, and DNA template 3 μL). The PCR conditions included initial denaturation (1 cycle, 95 °C/5 min); denaturation (30 cycles, 95 °C/30 s); annealing (30 cycles, 58 °C/30); elongation (30 cycles, 72 °C/30 s); final elongation (1 cycle, 72 °C/5 min); and cooling (1 cycle, 4 °C). The results were analysed using GelDoc 2000 software (BioRad, Feldkirchen, Austria) following the separation of the products in a 2% agarose gel at 100 V [24].

The susceptibility of the strains to 10 antibiotics was determined using a standard disc diffusion method on Mueller–Hinton agar plates (CM0337B, Oxoid, Basingstoke, Hampshire, UK), using a bacterial suspension with the turbidity adjusted to the 0.5 McFarland standard. The discs contained tetracycline 30 μg, neomycin 120 μg, enrofloxacin 30 μg, cefuroxime 5 μg, amoxicillin 10 μg, spectinomycin 100 μg, vancomycin 30 μg, tobramycin 10 μg, clindamycin 2 μg, and ciprofloxacin 5 μg. The plates were incubated in a thermostat at 37 °C for 24 h. Readings were based on the measurement of the diameter of the zone of bacterial growth inhibition within the antibiotic disc, which was compared with the reference values for a given antibiotic according to EUCAST [24].

### 2.3. Isolation and Characterization of Bacteriophages Specific for Isolated Staphylococcus *spp.* Strains

The bacteriophages specific for *Staphylococcus* spp. strains were isolated using samples obtained from the animal housing environments (floor, litter), as the natural conditions in which phages occur [25,26]. The bacteriophages were isolated using 18 h cultures of *Staphylococcus* spp. bacteria in TSB broth at pH 7.2 containing casein hydrolysate, soy hydrolysate, sodium chloride, potassium bicarbonate, and glucose. Each bacterial strain was suspended in a flask containing 20 mL TSB with 200 μL of 1 M MgCl_2_, to which 1 mL of filtrate of the sample from the floor, suspended in the TM buffer, was added. Then, the cultures were incubated in a water bath for 24 h at 37 °C. For each strain, a control flask containing the bacterial culture alone was prepared as well. The bacterial cultures in which cell lysis had taken place were then filtered using membrane filters with 0.45 µm and 0.22 µm pore size (MF-Millipore™ MembraneFilter, Merc-Poland, Warszawa, Poland).

The bacteriophages were purified on double-layer plates with TBS agar and tested for lytic activity. To this end, freshly prepared TSB agar (pH 7.3) was poured into sterile Petri dishes. The upper layer of the plate, in which the bacteria were embedded, consisted of 0.7% TSB agar in the amount of 5 mL per plate. Just before pouring the upper agar layer, 200 μL of 1 M MgCl_2_ solution, 1 M of CaCl_2_ solution, and 200 μL of bacterial broth culture specific for a given phage were added. The plates were incubated at room temperature for 20 min, after which 5 μL of the bacteriophage suspension was applied, followed by incubation at 37 °C for 24 h. Then, the plates were read for zones of inhibition of bacterial growth, known as plaques. The plaques were collected together with 0.7% agar and then suspended in TM buffer (50 mM Tris HCl, pH 7.5, 10 mM magnesium sulfate) and incubated at 37 °C for 4 h. Next, the suspension was centrifuged at 6000 rpm for 20 min in a centrifuge and filtered through 0.45 μm and 0.22 μm filters. The bacteriophage suspensions were stored at 4 °C for further analysis. For thorough purification of the bacteriophages, the procedure was carried out three times [27].

The morphological characterization of the bacteriophages was then carried out using Transmission Electron Microscopy (TEM) on slides negatively stained with 2% uranyl acetate [22,27]. Each slide was prepared using 5 μL of bacteriophage suspension in TM buffer. The morphological analysis was carried out at a magnification of 20,000–250,000 nm. The morphological classification of the phages was based on the structure and size of the capsid (e.g., helical, pleomorphic, icosahedral, filamentous/thread-like, or complex/polyhedral) and the structure and size of the tail [26,28].

The lytic titre of the bacteriophages was determined by the double-layer plate method developed by Marek et al. [27], using serial dilutions of the bacteriophages in TM buffer in a range from 10^−1^ to 10^−10^ PFU/mL, applied in the amount of 5 μL to the upper layer of the 0 7% TSB agar containing embedded cultures of *Staphylococcus* spp. strains specific for a given phage. The plates were then incubated for 24 h in a thermostat at 37 °C. The lytic titre was calculated after taking the dilution factor of the phage and the number of plaques and expressed in 1 mL into account. The phages were tested for stability in acidic and alkaline conditions (pH range 2–10) and higher temperature conditions (45 °C) according to methods proposed by Litt et al. [29]. The molecular analysis of the bacteriophages was based on isolated genetic material. The phage DNA was isolated using the commercial Norgen Biotek Phage DNA Isolation Kit (Norgen Biotek, Thorold, Canada) according to the method developed by the manufacturer. The qualitative analysis of the DNA obtained in this manner was performed by electrophoresis in 2% agarose using Quantity One 2000 software (BioRad), and quantitative analysis was carried out using the NanoDrop^TM^ Lite Spectrophotometer (Thermo Fischer Scientific). The DNA concentrations were expressed in ng/µL.

The restriction analysis of phage DNA was carried out using the restriction enzymes *EcoRI*, *NotI*, *BSU15I*, and *HindIII* (Invitrogen™ Anza™ 11 EcoRI, Invitrogen™ Anza™ 1 NotI, Invitrogen™ Anza™ 16 HindIII, and Invitrogen™ Anza™ 30 Bsu15I) according to the method provided by the manufacturer. Sixteen samples were prepared, containing 6 μL of DNA and 2 μL of an appropriate (according to the manufacturer’s recommendations) reaction buffer (ThermoScientific^TM^ Buffer Set for Restriction Enzymes). The restriction mixture was then incubated in a thermostat at 37 °C for 24 h, and restriction was carried out following the electrophoresis of the material in a 1% agarose gel stained with SimplySafe. The analysis was performed using Quantity One 2000 software (BioRad). The EurX 100–10,000 bp markers were used as standards.

### 2.4. Assessment of the In Vitro Antibacterial Effect of the Bacteriophages against Staphylococcus *spp.* Strains Isolated from Cattle

The experimental phage lysates were prepared from phages concentrated to titres of 2.2 × 10^−7^–3.2 × 10^−9^ PFU/mL in TM buffer in 50 mL Falcon tubes [22].

The antibacterial effectiveness of bacteriophages in vitro was assessed using the double-layer plate method with our own modification, which involved increasing the density of *Staphylococcus* spp. bacteria suspended in 0.7% TSB agar, and in many cases also using mixed bacterial cultures mimicking the natural environment. When the agar had set, 10 μL of the experimental preparation was applied to each plate at equal intervals, followed by incubation in a thermostat at 37 °C for 24 h.

Following incubation, the size, type, and purity of the bacterial growth inhibition zones were assessed, and on this basis the effectiveness of the phage preparations was determined. The lytic effects were compared with the lytic titres of the preparations, determined at the same time, to confirm their stability.

## 3. Results

### 3.1. Isolation and Identification of Bacteria

A total of 52 bacterial strains classified as *Staphylococcus* spp. were isolated from the 56 samples. Based on the morphological analysis, phenotypic properties, and antibiotic resistance profiles, only 16 strains, nos. 22557, 3324, 8534, 9228, 6036, 6019, 7248, 4299, 4296, 7559, 8597/L, 8597/P, 9602, 9779, 698, and 9398, assessed as pathogenic staphylococci uncontaminated by other microbes, including fungi and bacteria of the genus *Proteus*, were chosen for further analysis as potential hosts for phage isolation.

MALDI-TOF mass spectrometry identified 13 strains as *S. aureus* and 15 as *S. sciuri* at the level of highly probable genus identification and probable species identification. In addition, at the level of medium genus and species identification, 11 strains were identified as *S. aureus*, 13 as *S. sciuri*, and two as *Providencia stuartii* (Table 2).

Species identification of bacterial strains based on multiplex PCR confirmed the presence of genes specific for *Staphylococcus aureus* and *S. sciuri* (Figure 2).

### 3.2. Analysis of the Drug Resistance Profiles of Staphylococcus *spp.* Bacteria Isolated from Cases of Lameness in Dairy Cattle

The analysis of the drug resistance profiles of the bacterial isolates by the disc diffusion method revealed the presence of one strain showing resistance to eight of the 10 antibiotics tested and one strain resistant to six antibiotics. Six strains were simultaneously resistant to five antibiotics, three were resistant to four antibiotics, four were resistant to three antibiotics, and one was resistant to two antibiotics. Among the 10 antibiotics, representing different groups, none proved to be fully effective. Almost all isolates (94%; 15/16) showed resistance to neomycin, and more than 87% of strains (14/16) were resistant to spectinomycin. Only three strains showed intermediate susceptibility to enrofloxacin. Cefuroxime and vancomycin exhibited the highest antibacterial effectiveness (Table 3). In the case of vancomycin, we observed intermediate resistance; however, the results should be confirmed in a reference laboratory using MIC tests. Clindamycin also proved to be about 80% effective. Amoxicillin and tobramycin were 56.25% effective. None of the strains was susceptible to enrofloxacin.

The analysis of the results obtained in the PCR amplification of the bacterial strains confirmed the presence of products 533 bp in length, characteristic of the gene *mec*A, which determines resistance to methicillin, in more than half of the strains (9 of 16 strains had the *mec*A sequence in their genome). All strains subjected to restriction analysis showed a high concentration of DNA, from 23.5 ng/µL to 83.1 ng/µL.

### 3.3. Results of Isolation and Characterization of Bacteriophages Specific for S. aureus and S. sciuri Strains Isolated from Cattle with Signs of Lameness

Sixteen different bacteriophages specific for the isolated bacteria were obtained. Two of these were rejected due to poor lytic activity and lysogenic traits, based on analysis of plaques in the form of ring-shaped zones of bacterial growth inhibition on double-layer 0.7% LB agar plates. The remaining phages showed a wide spectrum of antibacterial activity, a wide host range, and a constant lytic titre. Three bacteriophages exhibited antibacterial activity against all 16 *Staphylococcus* strains (the bacterial host range column), four bacteriophages were capable of lysis against seven different bacterial isolates, three phages showed a lytic effect against four *Staphylococcus* strains, two showed antibacterial activity against three *Staphylococcus* spp. isolates, and four phages were capable of lysis against two *Staphylococcus* strains (Table 4).

Bacteriophages nos. 8597/L, 4296, and 698 exhibited a particularly wide range of antibacterial activity, comparable or sometimes even greater than that of the antibiotics tested in the study, resulting in very large clear zones of growth inhibition in the form of plaques around the sites where the phage preparations were applied.

It should be noted that an antibacterial effect was also observed against staphylococci at bacterial densities higher than the level recommended by EUCAST for antibiotic resistance tests in the disc diffusion method [24].

Examples of images of bacterial growth inhibition zones in the form of plaques on double-layer plates caused by the lytic effect of phages are presented in Figure 3.

Only three bacteriophages, nos. 8597/L, 4296, and 698, had fully stable titres in the pH range of 2–9 and in conditions of temperatures increased to 45 °C. In the case of other bacteriophages, lytic titres were significantly reduced, and the antibacterial effect obtained was negligible.

The morphological analysis of phages based on the results of electron microscopy (TEM) made it possible to distinguish bacteriophages containing morphological structures in the form of an icosahedral head and a tail, indicating that they belonged to the class *Caudoviricetes*, formerly *Caudovirales* (Turner et al. [30]). Six bacteriophages were assigned to the family of *Myoviridae*-like phages, based on icosahedral heads with sizes ranging from 64 to 95 nm and contractile tails from 180 to 200 nm in the extended state [22].

Nine bacteriophages were classified as the *Siphoviridae*-like family ICTV, [31] (Figure 4).

### 3.4. Restriction Analysis of Bacteriophages Specific for Staphylococcus *spp.* Using the Enzymes EcoRI, NotI, HindIII, and BSU15I

For the restriction analysis of the phages, genetic material was isolated in the form of DNA with concentrations within the range from 35.6 to 165.9 ng/µL.

The restriction analysis of the bacteriophages confirmed the presence of restriction sites for each of the four enzymes used, which made it possible to determine the characteristic restriction profile of each strain. In total, three restriction profiles were obtained for the 16 phages. The restriction profiles were similar for all phages after the digestion of phage DNA with *HindIII*, *EcoRI*, and *NOTI*. All of the bacteriophages were susceptible to digestion with *HINDIII*, *EcoRI*, and *NOTI*, but only four phages (φ 7559, φ 6019, φ 698, and φ 6019) were also susceptible to the enzyme BSUI51 (Figure 5).

### 3.5. Analysis of the Spectrum of Lytic Activity of the Three Bacteriophages with the Broadest Host Range

The assessment of the antibacterial effect of the three bacteriophages with the broadest spectrum of lytic activity against varied *Staphylococcus* species in the department’s collection (*S. aureus*, *S. sciuri*, and *S. epidermidis*), obtained from various animal species in previous research [22,24,32], confirmed the high antibacterial effectiveness of the phages, at levels from 86.1% to 100%. All of the bacteriophages exhibited lytic activity against 14 strains of *S. epidermidis*, and two exhibited 100% effectiveness against strains of *S. sciuri*, while phage φ 296 was most effective against *S. aureus* strains, causing lysis in 34 of 36 strains (Table 5).

## 4. Discussion

The present study, conducted in five herds of Holstein–Friesian cows, showed a high percentage of animals with clinical signs of lameness, amounting to about 70%. This study also confirmed the occurrence of pathogenic bacteria of the genus *Staphylococcus* isolated from the limbs of cows with lameness. The results are similar to data presented by other research centres regarding the percentages of animals with signs of lameness in a herd, and also confirm the types of microorganisms isolated from cases of hoof disease. Among these microbes, apart from anaerobes, special attention should be placed on the strains of *S. aureus* and *S. sciuri* [33,34]. The studies cited estimate the prevalence of lameness in breeding herds to be 40–50%, depending on the size of the herd. In practice, this means that every other cow has problems with proper movement and can be a potential source of infection for other animals. The present study confirmed that a substantial percentage (38%) of strains showed simultaneous resistance to five antibiotics. A disturbing finding was that none of the strains was susceptible to neomycin, and 93% of the strains were resistant to spectinomycin and enrofloxacin (81%). The results confirm those obtained by Monecke et al. [35], whose analysis of the drug resistance profiles of *S. aureus* bacteria isolated from cattle found a considerable percentage of strains (34 of 128 strains, 26%) to be resistant to the antibiotics tested.

In the present study, the genetic analysis of the drug resistance of *Staphylococcus* spp. strains showed the presence of the gene *mec*A, which determines resistance to methicillin, in more than half the strains tested (56%). This high percentage of strains with *mec*A confirms the results of a study by Patterson et al. [36], in which the genome of *Staphylococcus* spp. isolates had sequences of the gene *mec*A as well as the analogous gene *mec*C, which determine resistance to methicillin. The presence of the *mec*A gene among the strains has been confirmed at varied levels ranging from 0.7% to 90% [37,38].

Sixteen bacteriophages specific for the 16 isolated *Staphylococcus* strains were obtained. Based on the morphological analysis using TEM microscopy, seven of the phages were classified as *Myoviridae*-like and nine as *Siphoviridae*-like. This is being followed up with genetic analysis of the sequences, which will allow the phages to be assigned to new families according to the ICTV classification [30,31].

It is worth noting that in our study, the bacteriophages showing the broadest spectrum of lytic activity, against all tested bacterial strains of the species *S. aureus* and *S. sciuri*, were only found among *Myoviridae*-like phages. This made it possible to distinguish the three most valuable bacteriophages for further study to assess their antibacterial effect against 69 additional staphylococci obtained in previous research and belonging to the collection of the Department of Veterinary Prevention and Avian Diseases, including *S. aureus*, *S. sciuri*, and *S. epidermidis*. The phages showed a very broad spectrum of lytic activity, which is an argument in favour of considering their use to control cattle housing environments and infections caused by pathogenic strains of *Staphylococcus* spp.

It is also worth noting the limited stability of the titres of the phages, which after six weeks of storage significantly lost their antibacterial properties, with titres substantially reduced to 10^−2^ PFU/mL. In practice, this titre renders the phage unusable, due to the lack of an antibacterial effect. However, the present study confirmed the high stability of three of the phages in a wide pH range of 2–8 and a high temperature of 45 °C. The high stability of phage titres has also been confirmed by other researchers [39,40], who demonstrated that the use of phage therapy to control numerous infections is limited by the ability to store bacteriophages. In most cases, the lytic titre remains stable up to eight or even 12 weeks. The authors show that the stability of the lytic titres of phages is determined by their physicochemical traits, storage conditions, the type of stabilizer, and the properties of the packaging in which the phages are stored.

For example, a study investigating the effect of temperature on bacteriophage stability [39] showed that at low temperatures lytic activity declines rapidly without the use of protectants. Some strains even lost 96% of their lytic activity within 24 h of storage at 4 °C, while in the case of other strains (e.g., bacteriophage T4), the effect of temperature on activity seemed to be negligible. The effect of the pH of the environment on bacteriophages isolated from *S. aureus* cultures confirmed that bacteriophages can adapt to the specific conditions of pH 5.5, 6.5, 7.5, and 8.5 for up to three months. The results of these experiments have shown that an environment with pH close to neutral or slightly acidic is most favourable to the viability of phages and the stability of their lytic titre. Somewhat poorer results were obtained for an acidic and alkaline pH, in which the antibacterial effectiveness of the preparations fell to 70% and 80%, respectively [40].

The restriction profiles of the phages used in our study confirmed a genetic relationship between nearly all phages tested, except for four (φ 7559, φ 6019, φ 698, and φ 6019), which were additionally susceptible to enzyme BSUI51. However, the similarity in the restriction profiles of the phages did not translate to their host range and lytic activity. Other authors have shown similar restriction profiles among phages isolated from different environments [40,41]. Similar results were observed by Leite et al. [42], who also did not observe a correlation between the genetic relationships among six *S. aureus* phages, determined based on restriction analysis, and their lytic activity.

The positive results obtained by the researchers cited above and in the present study form the basis for developing therapeutic and preventive procedures using phage preparations to control and reduce bacterial infections. It is also worth noting that in the in vitro tests, a lytic effect was observed against 69 strains of *S. aureus*, *S. epidermidis*, and *S. sciuri* from our own department’s collection, obtained in previous research from various livestock farms, as well as against all 16 *Staphylococcus* strains which were hosts for the isolated bacteriophages. This confirms that these bacteriophages can potentially be used as a tool to control *Staphylococcus* infections in cattle, and above all, to improve the welfare of animals in their housing environments. The possibility of using bacteriophages successfully in experimental therapies in *Staphylococcus* infections in cattle has been confirmed in many in vitro studies and in in vivo studies using murine models [43,44].

The results are in agreement with those reported by Titze et al. [43], who used a cocktail of four phages, STA1.ST29, EB1.ST11, and EB1.ST27, with a titre of 1.2 × 10^9^ PFU/mL, specific for *Staphylococcus aureus*. That study showed high effectiveness in reducing *S. aureus* strains in milk samples, at a level of 86.6%, following eight hours of incubation. Another study [45] showed that two bacteriophages, *Staphylococcus* phage B4 and M8, exhibited a broad range of antibacterial activity against biofilm-forming methicillin-resistant (MRSA) strains of *S. aureus* isolated from cattle with mastitis.

The present study showed a broad spectrum of activity against pathogenic *Staphylococcus* spp. strains isolated from cattle with lesions in their limbs and hooves. The positive results indicate that the use of bacteriophages could be a helpful tool for improving the welfare of cattle in their housing environments.

It is also worth noting that the procedure developed in the present study enables simple, rapid (the isolation and purification of phage lysates takes no more than two days using only basic culture media, some filters, and buffers), and economical preparation of a phage product serving as an alternative to an antibiotic to control the presence of *Staphylococcus* spp. strains in livestock housing environments. Simple and economical methods of producing experimental phage preparations have also been presented in many published works [46,47].

The results of the present study also suggest that when developing an antibacterial formulation containing bacteriophages, we should first focus on choosing the right mixture of bacteriophages with stable lytic titres and a broad spectrum of antibacterial activity. Then, the stability of the product should be evaluated in in vitro conditions, and a treatment regime should be developed in field conditions. This is essential because, in many studies using bacteriophages, the results obtained in vitro have not translated to those obtained in in vivo conditions. Examples of such discrepancies include the attempts to develop experimental treatments for mastitis infections caused by staphylococci [48,49].

## 5. Conclusions

The high prevalence of multidrug resistant *Staphylococcus* spp. strains in dairy cow production, confirmed in the present study, poses a serious threat to people and the environment. The results suggest the need to look for alternative methods to control pathogens in livestock production.

Bacteriophages can be used in the future as an alternative tool, or to support current methods, for the treatment of both people and animals. Given the promising results of this preliminary research, which was the subject of the present study, such research should be continued in field conditions using phage cocktails with the broadest possible spectrum of lytic activity.

The positive results obtained in this study suggest that the use of bacteriophages can be one of the solutions for improving the welfare of cattle in their housing environments.

## Figures and Tables

**Figure 1 animals-14-00683-f001:**
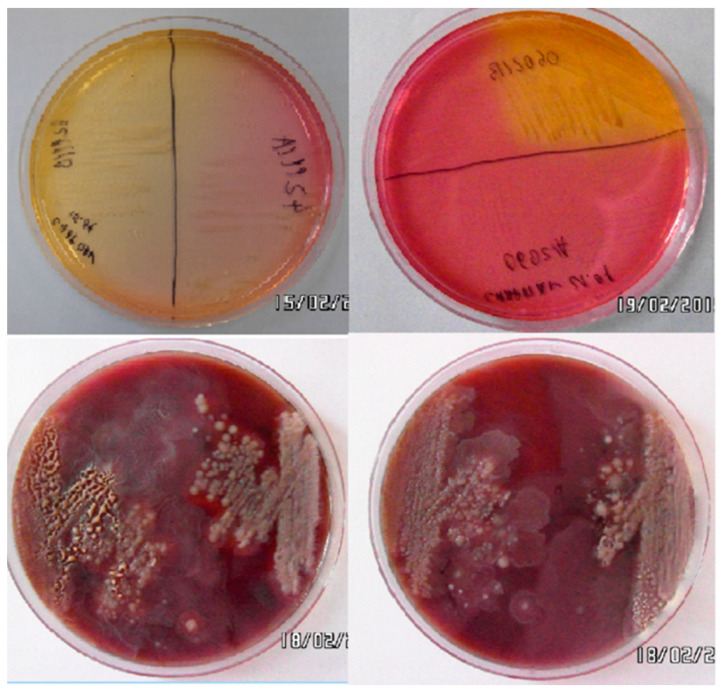
*Staphylococcus* spp. colonies grown on agar with 5% sheep blood (BIOCORP Columbia LAB-AGAR^TM^ + 5% KB) and Chapman differential media.

**Figure 2 animals-14-00683-f002:**
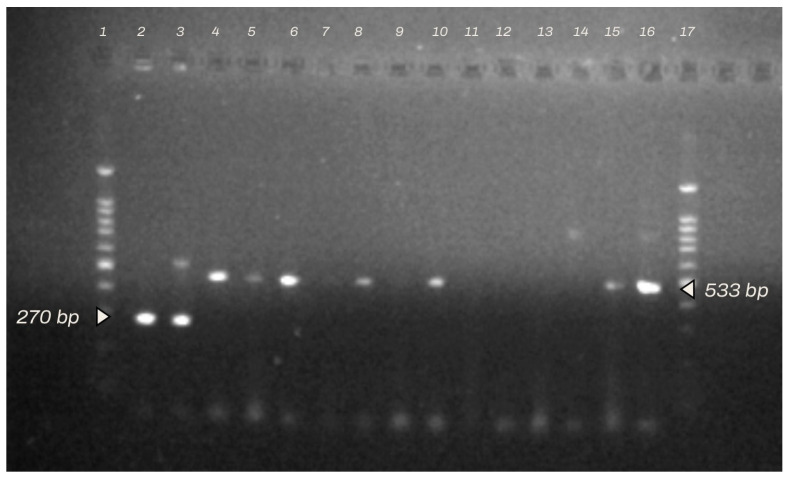
Electropherogram of multiplex PCR products of selected strains of *Staphylococcus* spp. (methicillin-resistant *S. aureus* and *S. sciuri*). Lanes 1, 1—Invitrogen^TM^ E-Gel 1 Kb Plus Express DNA Ladder (100–5000 bp), 2—strain 698, 3—strain 8597/L, 4—strain 8534, 5—strain 3324, 6—strain 6036, 7—strain 7559, 8—strain 4299, 9—strain 9779, 10—strain 6019, 11—strain 9602, 12—strain 8597/P, 13—strain 4296, 14—strain 7248, 15—strain-9228, 16—strain 9402, and 17—strain 22557. The presence of the genes *mec*A (533 bp) and *nuc* (270 bp).

**Figure 3 animals-14-00683-f003:**
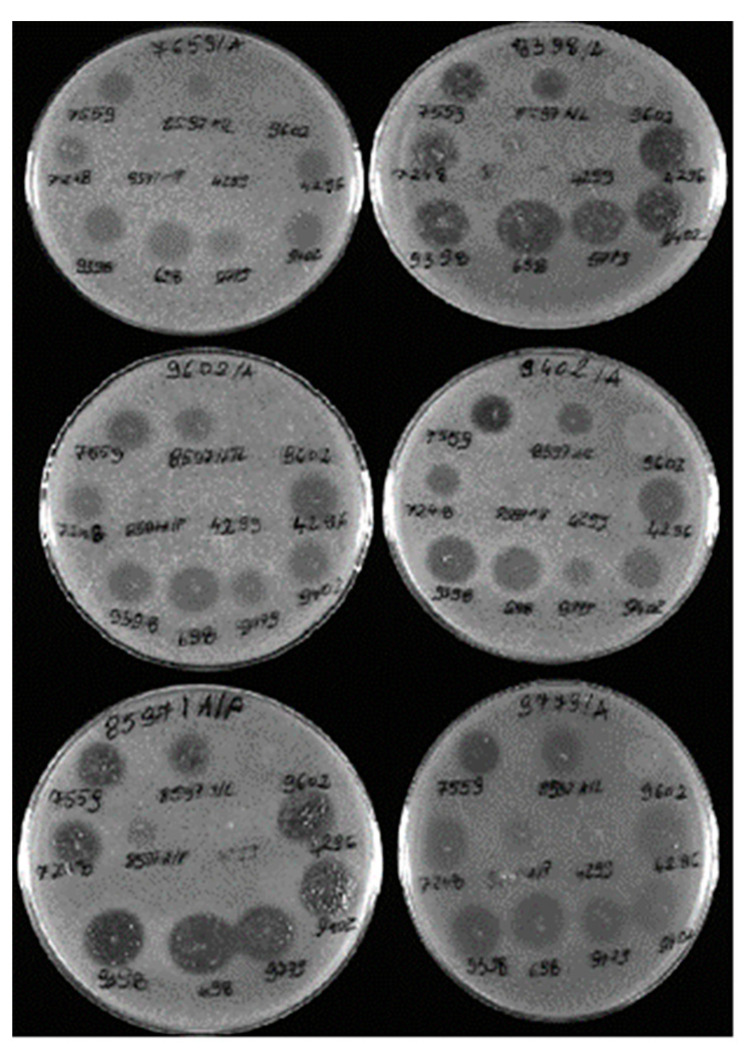
Examples of lytic zones (plaques) of bacteriophages specific for *Staphylococcus* spp. strains on double-layer TBS agar plates.

**Figure 4 animals-14-00683-f004:**
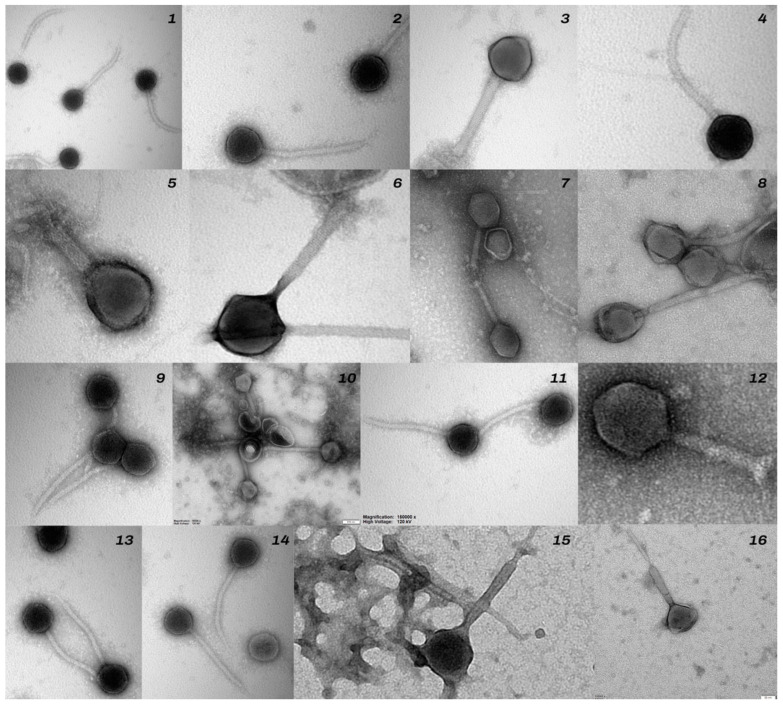
Negative-stained electron micrographs of phages specific for *Staphylococcus* spp. strains. 1—φ 7559; 2—φ 8597/P, 3—φ 8597/L, 4—φ 7428; 5—φ 4296; 6—φ 698; 7—φ 9398; 8—φ 9779; 9—φ 9402; 10—φ 9602; 11—φ 9228; 12—φ 8534; 13—φ 6019; 14—φ 6036; 15—φ 22557; 16—φ 3324.

**Figure 5 animals-14-00683-f005:**
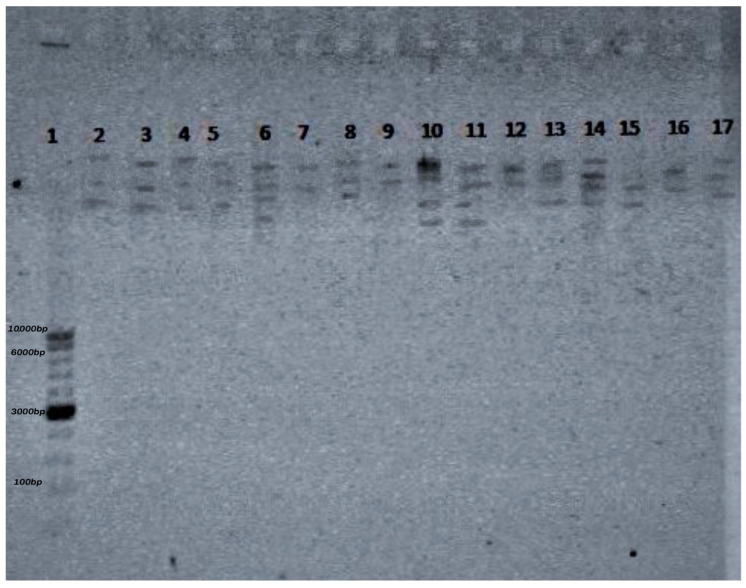
Restriction profile obtained following the digestion of the DNA of selected phages with restriction enzymes. 1—EurX Perfect 100–10,000 bp DNA Ladder, 2—φ 4296, enzyme *EcoRI*, 3—φ 7559, enzyme *EcoRI*, 4—φ 6019, enzyme *EcoRI*, 5—φ 698, enzyme *EcoRI*, 6—φ 4296, enzyme *HindIII*, 7—φ 7559, enzyme *HindIII*, 8—φ 6019, enzyme *HindIII*, 9—φ 698, enzyme *HindIII*, 10—φ 4296, enzyme *NotI*, 11—φ 7559, enzyme *NotI*, 12—φ 6019, enzyme *NotI*, 13—φ 698, enzyme *NotI*, 14—φ 4296, enzyme *BSU15I*, 15—φ 7559, enzyme *BSU15I*, 16—φ 7559, enzyme *BSU15I*, 17—φ 6019 enzyme *BSU15I*.

**Table 1 animals-14-00683-t001:** Detailed information about sample collection and number of cows.

Farm Number	Number of Samples Collected from Floor	Number of Cows from Which Samples Were Taken	Number of Cows with Two Injured Limbs	Total Number of Samples Collected from Cows
Farm 1	1	10	2	12
Farm 2	1	11	3	14
Farm 3	1	11	0	11
Farm 4	1	10	2	12
Farm 5	1	7	0	7
Total	5	49	7	56

**Table 2 animals-14-00683-t002:** Mean log(score) results for the MALDI-TOF MS analysis of bacteria isolated from dairy cows and their environment.

Log (Score)	Description	Symbol	Number of Strains		
			*Staphylococcus aureus*	*Staphylococcus sciuri*	*Providencia stuartii*
2.000–2.299	Secure genus identification and probable species identification	++	13	15	0
1.700–1.999	Probable genus identification	+	11	13	2
<1.700	No reliable identification	−	0	0	0

**Table 3 animals-14-00683-t003:** Antibiotic profiles of *Staphylococcus* spp. strains.

Strain No.	Antibiotic									
	TE	NEO	ENR	CXM	AMX	SPC	VA	TOB	DA	CIP
2257	S	R	R	S	S	R	S	S	S	S
3324	S	R	R	S	R	R	S	S	S	S
8534	I	R	R	S	R	R	I	S	R	S
9228	I	R	R	S	S	R	S	S	S	R
6036	I	R	R	S	S	R	S	S	I	S
6019	R	R	I	I	S	S	R	S	R	S
4296	S	R	R	S	I	R	I	S	R	R
7559	I	R	R	S	S	R	R	R	S	S
8597/L	I	R	R	S	S	S	S	R	R	R
8597/P	I	R	I	I	S	R	S	R	I	S
9602	R	R	R	S	I	R	S	R	I	R
9402	I	R	I	I	S	R	S	S	S	I
9779	I	R	R	S	R	R	S	S	S	R
698	I	R	R	S	R	R	S	R	S	I
3052	R	R	R	R	R	R	S	R	S	R
9400	I	S	R	S	S	R	I	I	R	I

TE—tetracycline 30 μg, NEO—neomycin 120 μg, ENR—enrofloxacin 30 μg, CXM—cefuroxime 5 μg, AX—amoxicillin 10 μg, SPC—spectinomycin 100 μg, VA—vancomycin 30 μg, TOB—tobramycin 10 μg, DA—clindamycin 2 μg, CIP—ciprofloxacin 5 μg, (S)—susceptible pathogen, (I) intermediate resistant pathogen, and (R) resistant pathogen.

**Table 4 animals-14-00683-t004:** Morphological characteristics of isolated bacteriophages, lytic titre, and host range.

	Phage No.	Bacterial Strain No.	Bacterial Species Used for Bacteriophage Isolation	Bacteriophage Family	Lytic Titre of Phage	Bacterial Host Range
1.	7559	7559	*S. sciuri*	*Siphoviridae*-like	1.8 × 10^−5^ PFU/mL	8597/P, 9402, 9779, 9398, 9602, 7559, 698
2.	8597/P	8597/P	*S. sciuri*	*Siphoviridae*-like	2.2 × 10^−8^ PFU/mL	8597/P, 7559, 9779
3.	8597/L	8597/L	*S. aureus*	*Myoviridae*-like	1.6 × 10^−7^ PFU/mL	8597/P, 8597/L, 9402, 7559, 9779, 4296, 9602, 9398, 22557, 6036, 3324, 9228, 6019, 3324, 8534, 698
4.	7248	7248	*S. sciuri*	*Siphoviridae*-like	1.9 × 10^−2^ PFU/mL	8597/P, 9402, 7559, 9779, 9602, 698, 3324
5.	4296	4296	*S. sciuri*	*Myoviridae*-like	1.8 × 10^−8^ PFU/mL	8597/P, 8597/L, 7559, 9402, 9779, 9602, 9398, 22557, 6036, 3324, 9228, 6019, 4296, 698, 4296, 698
6.	9398	698	*S. aureus*	*Myoviridae*-like	1.4 × 10^−5^ PFU/mL	8597/P, 8597/L, 4299, 9779, 4296, 9602, 9398, 22557, 6036, 3324, 9228, 6019, 7559, 3324, 698, 9402
7.	698	9398	*S. sciuri*	*Myoviridae*-like	1.2 × 10^−2^ PFU/mL	8597/P, 9402, 7559, 9779
8.	9779	9779	*S. aureus*	*Siphoviridae*-like	1.8 × 10^−1^ PFU/mL	8597/P, 4299, 7559, 4296
9.	9402	9402	*S. sciuri*	*Siphoviridae*-like	2.1 × 10^−3^ PFU/mL	8597/P, 7559, 9779, 9602
10.	9602	9602	*S. sciuri*	*Siphoviridae*-like	2.3 × 10^−1^ PFU/mL	9602, 3324
11.	9228	9228	*S. sciuri*	*Siphoviridae*-like	1.6 × 10^−2^ PFU/mL	22557, 6019, 6036, 9228, 8534, 698, 3324
12.	8534	8534	*S. sciuri*	*Myoviridae*-like	1.2 × 10^−2^ PFU/mL	8534, 3324, 6036
13.	6019	6019	*S. sciuri*	*Siphoviridae*-like	1.2 × 10^−4^ PFU/mL	3324, 9228
14.	6036	6036	*S. sciuri*	*Siphoviridae*-like	1.2 × 10^−4^ PFU/mL	3324, 22557
15.	22557	22557	*S. aureus*	*Myoviridae*-like	1.6 × 10^−2^ PFU/mL	3324, 22557, 9228, 6036, 698,8534, 3324
16.	3324	3324	*S. sciuri*	*Myoviridae*-like	1.2 × 10^−2^ PFU/mL	6036, 3324

**Table 5 animals-14-00683-t005:** Antibacterial activity of the three bacteriophages with the broadest host range against strains of three *Staphylococcus* species isolated from cattle and kept in the department’s collection.

Bacteriophage No.	Percentage (%) Lytic Activity against 69 *Staphylococcus* spp. Strains		
	*S. aureus* (*n* = 36)	*S. sciuri* (*n* = 19)	*S. epidermidis* (*n* = 14)
8597/L	88.9	100	100
4296	94.4	78.9	100
698	86.1	100	100

## Data Availability

Data are contained within the article.

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
