# Peer review of "Bacteriophages for Controlling Staphylococcus spp. Pathogens on Dairy Cattle Farms: In Vitro Assessment"

_animals, 2024, doi:10.3390/ani14050683_

Round 1

Reviewer 1 Report

Comments and Suggestions for Authors

The manuscript entitled “Assessment of the Potential Use of Bacteriophages to Control Staphylococcus spp. Pathogens in the Cattle Farming Environment based on Antibacterial Activity In Vitro” aimed to assess the antibacterial effectiveness of bacteriophages isolated from dairy cattle housing environments as potential tools in maintaining environmental homeostasis.

The introduction does not provide sufficient background. The first 50% (3 paragraphs) informs the general introduction to the problem of antibiotic use. Bacteriophages were introduced in the next paragraph for their replacement of antibiotics. Due to the study on staphylococci from lameness and bacteriophages from the environment, the authors need to describe both things and also make possible links. However, no information about those was described.

Description of materials and methods seems to be useful for microbiologists, not animal scientists. Even though, many descriptions are redundant. The following comments will be based on the work of animal scientists.

The first 3 paragraphs (L66-86) are about animal and sample collection. In animal science, it might be better to give information about animals and farms, so the reader can design to apply the results of the research to their situation. The authors need to inform the number of sample collections either one time or several times. How many cows are used for 49 samples collection? Why the samples of the environment were only 7 samples, please explain. By the way, most collected samples used in this study were related to environmental bacteria as the samples collecting method and culture were not specific for the diagnostic of lameness.

Minor comments

L79-81: delete. Due to the study did not make any relation to the severity of infection, it is not necessary to state here.

Methods of bacterial culture, identification and confirmation were described in 2.1 to 2.3.1 (L90-157). The description in this part was too detailed and redundant. Even in the work in animal science, the described information was too long. Please make it brief and make it in 1-2 paragraphs. Why do the authors do bacterial identification in many methods: multiplex PCR and MALDI-TOFF? Please explain. If it is not necessary, use only one.

Minor comments

L96-99: Delete.

Table 1 and Table 2: Delete and describe in text.

The authors produced bacteriophages specific for staphylococci from the samples obtained from animal housing environments (2.3.2). What is the reason for choosing these samples? Please explain these in the introduction part. It was a bit of detail, please refer to a previous study and shorten it.

In 2.3.3, please explain more about the analysis of the morphology of bacteriophages.

In 2.3.5, the authors stated to determine the molecular characterization of bacteriophages specific for Staphylococcus spp., it seemed to find the molecular character of bacteriophages. However, only the obtained DNA was analyzed. Please explain.

Why do the authors use the experimental phage lysates with the specified concentration? What are the references?

From 3.1, how do the authors determine the pathogenic strains since the author used selective media and also collecting samples from the skin was always difficult to avoid contamination? Please clarify.

Morphological analysis (gram stain) of microscope images with Figure 3 (L252-256) is not necessary, please delete.

L264-272 (also Figure 4) is not clear. Also, it is redundant to MALDI-TOF for bacterial species confirmation. Please consider deleting it.

L275-277: Analysis of the drug….to six antibi

3.2 and Table 4: The interpretation of antibiotic resistance has to be reported as S, I or R depending on the EU cast, please revise both text and Table. It is also better to summarize into groups of bacteria that have the same pattern of antibiotic resistance.

L295-298 “Analysis of the results obtained in PCR amplification of the bacterial strains confirmed the presence of products 750 bp in length, characteristic of the gene mecA, which determines resistance to methicillin”. Please clarify how you confirmed the products' 750bp in length. However, the references to the gene mecA must be explained in M&M, unless otherwise referred it in the discussion part.

Table 5 is too much detail, please delete it.

L314-320 must be in the M&M part.

Figure 5 is not clear.

L356-331: Please explain how to classify phage types in M&M.

Table 7 is too much detailed, please delete it.

L340 to 353 and Figure 7: Please use the better image figure and explain the relationship between the restriction site or enzyme and phages clearly.

L379-389: It is a bit difficult to state that because your study did not identify other bacteria genera or species.

L394-402 Do you have any confirmation method to confirm that the finding product 7500 bp in length is the gene mecA? Please clarify.

L426-436: The effect of temperature and pH on bacteriophage stability was not explored in this study. Please delete it.

L442-444: It is too early to conclude because many experiments need to be confirmed. Please revise it.

Many sentences in the discussion are exaggerated over the findings of this study. Please revise those.

Author Response

Thank You very much for the revision of the manuscript and for all suggestion and comments. Thank You for all additional informations and questions. We really appreciate Your work and help. According to the suggestions we have made the proper corrections. We also tried to answer to all questions. We hope that the manuscript will be accepted for publication in it’s current form (see comments below in blue colour).

Once again

Thank You very much

The authors

Reviewer 2 Report

Comments and Suggestions for Authors

General comments and suggestions

The study aimed to contribute to controlling infectious diseases on dairy cattle farms by using bacteriophages as an alternative to antibiotics. It is a good study, given the increasing challenges of antimicrobial resistance. However, bacteriophages have several limitations, including instability in hostile environments and destruction by UV light and chemicals. How are you going to overcome this?

Authors should write the conclusion based on the findings of the present study. The authors conclude that bacteriophages can be used by adding them to disinfectants. Did you perform such an experiment? The findings would have been a new contribution to dairy cattle farming concerning controlling infectious diseases. Where are the results?

Authors need to rewrite the manuscript with well-structured paragraphs as the current form. Some of the paragraphs are just one sentence, e.g., lines 45 – 46 and 457 – 459, to mention a few, and in most cases, they don't lead to the next.

Throughout the manuscript, replace "symptoms" with "signs", which is the correct term in veterinary medicine/science.

The manuscript misses key sections such as simple summary, author contributions, funding, institutional review board statement, informed consent statement, data availability Statement, acknowledgements, and conflicts of interest. Please add these sections.

Add more references, as currently, there are only 29!

Specific comments

The title is too long; consider making it more specific to your study. I suggested "Bacteriophages for Controlling Staphylococcus spp. Pathogens on Dairy Cattle Farms: In vitro Assessment".

Add a simple summary.

An abstract is too extended (more than 200 words).

Introduction

Lines 39 – 40 add references.

Lines 41 – 42 add references.

Materials and Methods

Add information about the location where the study was conducted.

Did you have animal ethics clearance for your study?

Lines 72 – 74, these are the results?

Figures 1 and 2 might be unnecessary.

Results

This section needs to be clearly presented. It needs to be made clear how many samples were collected from animals and how many samples were collected from the floors—too many Tables and Figures.

Line 19 contradicts line 245 regarding samples.

 Discussion

This section needs to be restructured with clear paragraphs.

Lines 380 and 384 replace symptoms with clinical signs.

Lines 457 – 459: Is your procedure simple and rapid? How many days did you use to isolate bacteriophages, let alone the cost of reagents?

In this section, you need to provide the limitations of your study.

 Conclusion

The conclusion needs to be more precise and supported by the current results rather than previous studies.

Lines 472 – 473: Repetition of the introduction; the statement fits well in the introduction section. Can you please consider moving the statement into the introduction section with references? 

Lines 477 – 478: Where are you drawing this conclusion from? Did you add bacteriophages to disinfectants and test them?

Comments on the Quality of English Language

Check on grammar, structure, and paragraph.

Author Response

Reviewer 2

Dear Reviewer

Thank You very much for the revision of the manuscript and for all suggestion and comments. Thank You for all additional informations and questions. We really appreciate Your work and help. According to the suggestions we have made the proper corrections. We also tried to answer all the questions asked. We hope that the manuscript will be accepted for publication in it’s current form (see comments below in blue colour).

Once again

Thank You very much

 The authors

General comments and suggestions

The study aimed to contribute to controlling infectious diseases on dairy cattle farms by using bacteriophages as an alternative to antibiotics. It is a good study, given the increasing challenges of antimicrobial resistance. However, bacteriophages have several limitations, including instability in hostile environments and destruction by UV light and chemicals. How are you going to overcome this?

A: The phages were tested about stability for acidic and alkaline conditions pH range from 2-10. We have chosen only this phages which express the stability between pH 3-8 (this is pH of most used dissinfectants). We also added this information in the methods chapter with proper references.

Authors should write the conclusion based on the findings of the present study. The authors conclude that bacteriophages can be used by adding them to disinfectants. Did you perform such an experiment? The findings would have been a new contribution to dairy cattle farming concerning controlling infectious diseases. Where are the results?

A:The conclusion chapter has been rewritten according to the reviewers comments. We have used the bacteriophages as supplements of disinfectants in poultry production. However the results of these study were not yet published. So we decided to not write about this proposal in this manuscript. Thank You very much for Your comments we really appreciate Your help.

Authors need to rewrite the manuscript with well-structured paragraphs as the current form. Some of the paragraphs are just one sentence, e.g., lines 45 – 46 and 457 – 459, to mention a few, and in most cases, they don't lead to the next.

A: We have corrected these paragraphs by adding additional information with proper cited references or by combinig the sentences. Thank You very much for Your suggestion.

Throughout the manuscript, replace "symptoms" with "signs", which is the correct term in veterinary medicine/science.

A: This correction has been made according to the suggestion. Thank You very much for Your comment.

The manuscript misses key sections such as simple summary, author contributions, funding, institutional review board statement, informed consent statement, data availability Statement, acknowledgements, and conflicts of interest. Please add these sections.

A: This lacking key sections has been added according to the editor’s and rewievers comments. Thank You very much for Your help.

Add more references, as currently, there are only 29!- The proper references has been added and for better expression the added references are in blue colours.

Specific comments

The title is too long; consider making it more specific to your study. I suggested "Bacteriophages for Controlling Staphylococcus spp. Pathogens on Dairy Cattle Farms: In vitro Assessment".

A: The title has been corrected according to the instructions. Thank You very much for Your help we really appreciate Your work.

Add a simple summary.- It was added according to the suggestions.

An abstract is too extended (more than 200 words). It was corrected.

Introduction

Lines 39 – 40 add references.-

Lines 41 – 42 add references.

A: The proper references has been added according to the suggestions. Thank You very much for Your help.

Materials and Methods

Add information about the location where the study was conducted.- It was added according to the suggestions

Did you have animal ethics clearance for your study?

A: In this study we used we used a standard veterinary services and  veterinary clinical trials, so according to the main Polish legislation act: ACT of January 15, 2015 on the protection of animals used for scientific or educational purposes chapter 1 art. 1  (*Dz.U. 2015 poz. 266) we didn’t have to obtain the proper ethical agreement.- The proper information were added at the end of the text of manuscript.

Lines 72 – 74, these are the results?

A: This sentence has been corrected in formula of material and methods.

Figures 1 and 2 might be unnecessary.

A:Thank You very much for your suggestions. We only would like to show the cases of lameness in cows- were we collect the samples. However we can remove this Figure. In case of Fig, 2 we would like to show the morphological confirmation of Staphylococcus spp. strain. So if it’s possible we would like to leave this figure in the text of manuscript.

Results

This section needs to be clearly presented. It needs to be made clear how many samples were collected from animals and how many samples were collected from the floors—too many Tables and Figures.-

A: We have corrected this chapter oslo by removing of three figures and four tables.

Line 19 contradicts line 245 regarding samples.- This fragment was corrected.

Discussion

This section needs to be restructured with clear paragraphs.

Lines 380 and 384 replace symptoms with clinical signs.

Lines 457 – 459: Is your procedure simple and rapid? How many days did you use to isolate bacteriophages, let alone the cost of reagents?

In this section, you need to provide the limitations of your study.

A: This section has been corrected according to cited points.  We have added some examples of references to confirm the economical aspect of using bacteriophages in phage tehrapies. Thank You very much for Your suggestions we really appreciate Your help.

 Conclusion

The conclusion needs to be more precise and supported by the current results rather than previous studies.

Lines 472 – 473: Repetition of the introduction; the statement fits well in the introduction section. Can you please consider moving the statement into the introduction section with references?

A: This sentence has been corrected according to the instruction. We have transferred some part of the sentence to the introduction with proper references.

 Lines 477 – 478: Where are you drawing this conclusion from? Did you add bacteriophages to disinfectants and test them?-

A: This part of the manuscript has been  corrected and we added the proper additional information. Some part of the text has been removed. The detailed explanation was also added as answers in Genera comments. Thank You very much for Your work.

Comments on the Quality of English Language Check on grammar, structure, and paragraph.

A: This manuscript after revision have been sed once again to the native speaker for English correction. We hope that in this form the English Language is more quality. Thank You very much for Your comments.

Reviewer 3 Report

Comments and Suggestions for Authors

It is an interesting idea and concept. However I am worried about the specific aim of the study, and how the laboratory science is applied to the clinical aspects of dairy cattle.

You have said that you are aiming to reduce the environmental of coagulase positive Staphs which cause disease in dairy cattle. However the main disease coagulase positive Staphs cause in dairy cattle is mastitis/high SCC, which is spread contagiously from cow to cow in the milking parlour, not via the environment. You also then discuss lameness; however Staphylococcus species are not a major cause of dairy cattle lameness -that is Treponemes and interdigital necrobacillosis; neither of which are Staphs or caused by Staphs. Therefore I am struggling to work out how the bacteriophages are going to have a clinical application. We also know that Staphs are present as part of the normal microbiome on cattle, thus by removing them, we may alter the microbiome in a negative manner.

There are also issues surrounding what you are defining as clinically lame, as well as the prevalence of lameness you have reported (in the conclusions, not the results) and as to how you have determined the prevalence of lameness. The focus of the discussuion is on lameness, which is not focussed upon in the introduction. There also seems to be this idea that the presence of pathogenic bacteria in a sample and environment means it is the cause of disease; cryptosporidium, rotavirus, coronavirus, E.coli etc are all found in cattle faeces; this does not mean they are causing disease in these adult cattle. Similarly respiratory bacteria are commensals in the throat, but are pathogenic in the lungs - be careful around correlation becoming causation and an issue. 

It would also be good to have sections in both of the introduction and discussion on bacteriophages, how they work and how they have previously been utilised in the field of bovine health and management. The introduction in general needs to be more specific (what is the prevalence of Staph mastitis? economic impact? welfare impact? risk of culling) as it does not allude to the problem which Staphs cause if present. 

It was a good idea but I don't think it has been linked in the correct clinical context, sorry.

Comments on the Quality of English Language

The English is good; just try to keep to scientific words such as treatment rather than fight infection

Author Response

Reviewer 3

Thank You very much for the revision of the manuscript and for all suggestions and comments. Thank You for all additional informations and questions. We really appreciate Your work and help. According to the suggestions we have made the proper corrections. We also tried to answer all the questions. We hope that the manuscript will be accepted for publication in it’s current form (see comments below in blue colour).

Once again

Thank You very much

 The authors

It is an interesting idea and concept. However I am worried about the specific aim of the study, and how the laboratory science is applied to the clinical aspects of dairy cattle.

A: Thank You very much for Your question. We really appreciate Your work and help.The main goal of the research and its implementation will allow the development of new and ecological activities in the field of controlling the prevalence of pathogens in animal housing environments, including: by improving their well-being and the well-being of the environment.

You have said that you are aiming to reduce the environmental of coagulase positive Staphs which cause disease in dairy cattle. However the main disease coagulase positive Staphs cause in dairy cattle is mastitis/high SCC, which is spread contagiously from cow to cow in the milking parlour, not via the environment. You also then discuss lameness; however Staphylococcus species are not a major cause of dairy cattle lameness -that is Treponemes and interdigital necrobacillosis; neither of which are Staphs or caused by Staphs. Therefore I am struggling to work out how the bacteriophages are going to have a clinical application. We also know that Staphs are present as part of the normal microbiome on cattle, thus by removing them, we may alter the microbiome in a negative manner.

A: Thank You very much for Your suggestions and for question. We decided to use the Staphylococcus strains because of the most often prevalence of this bacteria in cattle environment especially is most often isolated from injures of hooves and limbs. According to some research this bacteria is capable of enhancing the hooves inflammatory process and is most often isolated in these lesions in cows.  We completly agree with You than F.necrophorum is the predominat agent in DD infections, however F. necrophorum has a strong synergic cooperation with other bacteria including  Staphylococcus  or Escherichia coli in causing infections in cows. We have added proper information in the introduction chapter of the manuscrip with proper references.

See also some examples in articles below

Simpson , K.M.; Streeter , R.N., Jones, M.L.; Taylor, J.D.; Callan, R.J.; Holt, T.N.;  Review of digital anatomy, infectious causes of lameness, and regional intravenous perfusion in cattle. The Bovine Practitioner, 2020, 54, 17-29.

Biggs et al. Foot Rot in Cattle https://extension.okstate.edu/fact-sheets/foot-rot-in-cattle.html

There are also issues surrounding what you are defining as clinically lame, as well as the prevalence of lameness you have reported (in the conclusions, not the results) and as to how you have determined the prevalence of lameness. The focus of the discussuion is on lameness, which is not focussed upon in the introduction. There also seems to be this idea that the presence of pathogenic bacteria in a sample and environment means it is the cause of disease; cryptosporidium, rotavirus, coronavirus, E.coli etc are all found in cattle faeces; this does not mean they are causing disease in these adult cattle. Similarly respiratory bacteria are commensals in the throat, but are pathogenic in the lungs - be careful around correlation becoming causation and an issue.

It would also be good to have sections in both of the introduction and discussion on bacteriophages, how they work and how they have previously been utilised in the field of bovine health and management. The introduction in general needs to be more specific (what is the prevalence of Staph mastitis? economic impact? welfare impact? risk of culling) as it does not allude to the problem which Staphs cause if present.

A: We have made a proper corrections and add additive paragraphs with bacteriopahes as well as with prevalence and economic problem of S. infections accoridng to the suggestions. We have also added the proper explanation in additional sentences about the role of Staphylococcus spp in  lameness development. Thank You very much for Your comments.

It was a good idea but I don't think it has been linked in the correct clinical context, sorry.- Thank You very much for Your comment. We have made some additional fragments to create the basic and more clinical aspects of the study.

Comments on the Quality of English Language

The English is good; just try to keep to scientific words such as treatment rather than fight infection

A: The manuscript has been once again corrected by native speaker for increase of quality of English language. Thank You very much for Your comments.

Round 2

Reviewer 1 Report

Comments and Suggestions for Authors

Thank you for the improvement of the manuscript based on the suggestions. I still have a few comments to improve the manuscript. 

Reviewer 1

The manuscript entitled “Assessment of the Potential Use of Bacteriophages to Control Staphylococcus spp. Pathogens in the Cattle Farming Environment based on Antibacterial Activity In Vitro” aimed to assess the antibacterial effectiveness of bacteriophages isolated from dairy cattle housing environments as potential tools in maintaining environmental homeostasis.

The introduction does not provide sufficient background. The first 50% (3 paragraphs) informs the general introduction to the problem of antibiotic use. Bacteriophages were introduced in the next paragraph for their replacement of antibiotics. Due to the study on staphylococci from lameness and bacteriophages from the environment, the authors need to describe both things and also make possible links. However, no information about those was described.

A: We have add the additionary paragraph in introduction chapter with information about the bacteriophages and using in therapy in humans and animals (see the proper references cited in the text of manuscript). Thank You very much for Your suggestions.

V2 comment: Thank you for adding more information of bacteriophages (Line 102-115) to the comments, but it is a bit too long. The comment on ‘The first 50% (3 paragraphs) informs the general introduction to the problem of antibiotic use” intended the author to shorten this introduction. This manuscript is not related to mastitis. Please delete unnecessary information about mastitis in this manuscript.

Line 64-68: Delete

Line 69-78: Delete (due to it is not related to the main objective of this manuscript.

Line 80-81: Delete a sentence “According to …”.

Line 86: add ref 12 with 11 and delete the next sentence.

Line 92-97: Delete as this manuscript was not related to mastitis, and this confuses readers.

Line 106-107: Delete a sentence “ This means…”

Line 109-113: Delete a sentence “Bacteriophages were first….1917”

Description of materials and methods seems to be useful for microbiologists, not animal scientists. Even though, many descriptions are redundant. The following comments will be based on the work of animal scientists.

The first 3 paragraphs (L66-86) are about animal and sample collection. In animal science, it might be better to give information about animals and farms, so the reader can design to apply the results of the research to their situation. The authors need to inform the number of sample collections either one time or several times. How many cows are used for 49 samples collection? Why the samples of the environment were only 7 samples, please explain. By the way, most collected samples used in this study were related to environmental bacteria as the samples collecting method and culture were not specific for the diagnostic of lameness.

A: We have add the proper information about the samples collection and cows according to the suggestions in material and methods chapter. We have also reworded individual sentences in this paragraph to make it clear that this is a "material and methods" chapter, in line with the reviewers' suggestions. We hope that in this form is much better understable for the reader. In the case of samples taken from animal housing environments (floors, bedding - if any), there was an error in the manuscript, for which we sincerely apologize. There were five samples because one sample for phage isolation came from each farm. Thank You very much for Your comments we really appreciate Your help.

V2: Thank you for this revised version. Based on many animal research, information on farm and farm sample collection should be described first, and then cow information and cow sample collection will be explained later. Please correct some points:

Please revise Lines 129-142.

English must be checked.

Line 28: Change to ‘Materials”

 Many sentences in the discussion are exaggerated over the findings of this study. Please revise those.

A: We have made a proper corrections according to the suggestions. We added additionally sentences according to the Reviewers comments. We hope that in this form the discussion chapter is better understable. One again thank You very much for Your comment and help.

Once again thank You very much for Your help and assistance.

 V3: This study collected samples from 5 farms, so the overestimate on epidemiology should be deleted.

Lines 484-496: Delete

Comments on the Quality of English Language

It would be better to have an English check. I found several small mistakes.

Author Response

Thank You very much for the revision of manuscript ad for all comments. We really appreciate Your work and help.

We have made a proper corrections according to the suggestions and proposed instructions.

Because there were too many corrections in the manuscript after previous proofreading, we decided to accept the previous changes in the text and the changes currently introduced apply only to the stage 2 reviews.

We also added the answers to the comments which are marked in blue colour.

We hope that in its current form the manuscript could be acceptable for publication.

Once again thank you very much for Your assistance.

Reviewer 2 Report

Comments and Suggestions for Authors

General comments and suggestions

The authors have addressed some of the previous comments and suggestions. They should consider the following additional comments and suggestions to improve the manuscript.

The authors need to clarify how many samples were collected from each of those five farms (litter and floor) for bacteriophage isolation.

The results section needs to be presented clearly and only the results/findings of the present study, not the previous research.

Specific comments

Lines 23 – 24: Please be consistent with how you are reporting your findings. If you use percentage and fraction, then stick to it. It has been better presented in the results section, lines 365 – 367.

Lines 27 – 28: The word bacteriophage is missing – consider adding it, please.

I wonder if this is simple as complex/scientific words are used in this section. In this section, abbreviations are not required - consider using full words, please. 

Introduction

Line 64: Limit to three most relevant references.

Lines 67 – 68 add references.

Line 75: Others – which ones? It is better to list them.

Lines 82 – 85: Add references.

Lines 102 – 103: Add references.

Lines 103 - 106: Add references.

Materials and Methods

Line 137: After the Lublin region, add the country, please.

Lines 284 – 285: Consider removing the heading as it is under the 2.3.

How many samples were collected from each farm (litter and floor) for bacteriophage isolation is unclear. Clarify this, please. You could tabulate this information farms (1,2,3,4 and 5) against samples (litters and floors).

Results

Line 329: Please double-check if it is 46 samples or 56 samples.

Lines 335 – 337: It is basic information so unnecessary as it is for the Figure 3 you deleted.

Lines 412 – 417: The description is not clear, and Table 3 is not clear too.

Lines 425 – 426: Please add a reference (s) to qualify your statement.

Figure 3 is unnecessary if you prefer keeping it – consider supplying a large and clear image.

Figures 2 – 5 might fit better as appendices. Consider them, please.

Lines 473 – 473: Section 3.5 – I am confused and hope other readers will be too.

Was what you call “the department collections” part of this study? You have not mentioned it before. It would be nice to stick only to the present study. In this section, you can only report the phage activity against 16 Staphylococci isolates that were used for further analysis or 52, the total number of Staphylococci isolated during the present study.

Discussion

Lines 531 – 534: This was not part of the study – Check the same comments in the results section.

Also, check lines 573 – 576 for the same comment.

Conclusion

Lines 614 – 622: Streamline to make the statements precise.

Author Response

(The authors gave the same response as above.)
